# Design principles of the ESCRT-III Vps24-Vps2 module

Sudeep Banjade[1,2], Yousuf H Shah[1,2], Shaogeng Tang[3], Scott D Emr[1,2]*

[1]Weill Institute for Cell and Molecular Biology, Cornell University, Ithaca, United States; [2]Department of Molecular Biology and Genetics, Cornell University, Ithaca, United States; [3]Department of Biochemistry, Stanford University, Stanford, United States

**Abstract** ESCRT-III polymerization is required for all endosomal sorting complex required for transport (ESCRT)-dependent events in the cell. However, the relative contributions of the eight ESCRT-III subunits differ between each process. The minimal features of ESCRT-III proteins necessary for function and the role for the multiple ESCRT-III subunits remain unclear. To identify essential features of ESCRT-III subunits, we previously studied the polymerization mechanisms of two ESCRT-III subunits Snf7 and Vps24, identifying the association of the helix-4 region of Snf7 with the helix-1 region of Vps24 (Banjade et al., 2019a). Here, we find that mutations in the helix-1 region of another ESCRT-III subunit Vps2 can functionally replace Vps24 in *Saccharomyces cerevisiae*. Engineering and genetic selections revealed the required features of both subunits. Our data allow us to propose three minimal features required for ESCRT-III function – spiral formation, lateral association of the spirals through heteropolymerization, and binding to the AAA + ATPase Vps4 for dynamic remodeling.

*For correspondence:
sde26@cornell.edu

Competing interests: The authors declare that no competing interests exist.

## Introduction

ESCRTs (endosomal sorting complexes required for transport) control a growing list of membrane remodeling events in cells (*Vietri et al., 2020*). Among the different subcomplexes of ESCRTs, ESCRT-III is required in all processes, while the requirement of the other upstream complexes 0, I, and II is variable (*Henne et al., 2013*). Our understanding of the mechanisms of ESCRT-III assembly has increased substantially in the last decade (*Henne et al., 2012*; *Shen et al., 2014*; *Cashikar et al., 2014*; *Chiaruttini et al., 2015*; *McCullough et al., 2015*; *Tang et al., 2015*; *Adell et al., 2017*; *Mierzwa et al., 2017*; *Schöneberg et al., 2018*; *Goliand et al., 2018*; *Maity et al., 2019*; *Bertin et al., 2020*; *Moser von Filseck et al., 2020*; *Nguyen et al., 2020*; *Pfitzner et al., 2020*). However, important questions remain. The specific role of each of the eight ESCRT-III subunits remains unclear. In eukaryotes, eight ESCRT-III proteins exist: Did2 (CHMP1), Vps2 (CHMP2), Vps24 (CHMP3), Snf7 (CHMP4), Vps60 (CHMP5), Vps20 (CHMP6), Chm7 (CHMP7), and Ist1 (IST1) (*Rue et al., 2008*; *McCullough et al., 2018*). Of these, Vps20, Snf7, Vps24, and Vps2 have been studied in greater detail, because they are individually essential for multivesicular body (MVB) biogenesis in yeast, the earliest ascribed role for ESCRTs. These four proteins are the minimal subunits necessary to create intraluminal vesicles in the MVB pathway, as suggested by in vivo and in vitro analyses (*Adell et al., 2017*; *Schöneberg et al., 2018*; *Pfitzner et al., 2020*; *Teis et al., 2008*). A recent study has further included Did2 and Ist1, providing additional insights into the sequential recruitment mechanism of ESCRT-III components (*Pfitzner et al., 2020*). Toward a comprehensive understanding of each ESCRT-III subunit, studies in MVB biogenesis have provided important clues regarding their individual roles in membrane remodeling.

Vps24 and Vps2 are essential for MVB biogenesis (*Henne et al., 2012*; *Babst et al., 2002*; *Buchkovich et al., 2013*). Vps24 and Vps2 are also recruited cooperatively to membranes, requiring

each one for the other's efficient recruitment (*Adell et al., 2017*; *Mierzwa et al., 2017*; *Babst et al., 2002*; *Banjade et al., 2019a*). Because of this reason, these two proteins have been analyzed together in previous ESCRT-related work. Interestingly, in previous work, it was found that during HIV budding, while CHMP2 (the human ortholog of Vps2) is essential, CHMP3 (the human Vps24 ortholog) is not essential for HIV egress from cells (*Morita et al., 2011*). We hypothesized that there may be features of Vps24 and Vps2 that render CHMP3 non-essential in some ESCRT-dependent processes. What are these essential features in these two proteins that make them indispensable for membrane remodeling? We set out to define those features in this work.

One of the most important features of Vps2 that has previously been described is the recruitment of the AAA + ATPase Vps4, which modifies ESCRT-III polymers (*Obita et al., 2007*; *Adell et al., 2014*). Vps24 and Vps2 bind to Snf7 and remodel Snf7 polymerization, changing the flat Snf7 spiral into three-dimensional (3D) helices (*Henne et al., 2012*; *Moser von Filseck et al., 2020*; *Pfitzner et al., 2020*; *Banjade et al., 2019a*). We hypothesized that a single protein with features of Vps24 and Vps2 that can bind to Snf7 but also recruit Vps4 may be sufficient for function. Here, through engineering approaches, we define a single protein that possesses such functions of both Vps24 and Vps2. These data allow us to define the minimal essential properties of an ESCRT-III heteropolymer that are required for intraluminal vesicle formation. These include (a) spiral formation through a Snf7-like molecule, (b) lateral association through a Vps24/Vps2-like molecule, and (c) the ability to recruit the AAA + ATPase Vps4.

## Results

### Overexpressing Vps2 can replace the function of Vps24 in MVB sorting

In our previous work (*Banjade et al., 2019a*), we observed that overexpressing *VPS24* suppresses the defect of a *snf7* allele (*snf7-D131K*) that encodes a Snf7 mutant with a lower affinity to Vps24. The overexpression, however, does not rescue other *snf7* alleles that encode defective Snf7 homopolymers. We also observed that overexpression of Vps2 rescues the defect of *snf7-D131K*. These data are consistent with the observations that Vps24 and Vps2 bind synergistically to Snf7 (*Adell et al., 2017*; *Mierzwa et al., 2017*; *Babst et al., 2002*). Following these observations, we sought to test whether expressing a high level of Vps2 also rescued the lack of Vps24 in cells, with the hypothesis that Vps2 may possess a lower affinity binding surface for Snf7, which could be overcome with an increased availability of Vps2 in the cytoplasm. These ideas are also consistent with previous binding-constant measurements with mammalian Vps2 (CHMP2A) and Vps24 (CHMP3), which showed that CHMP3 possesses a 16-fold tighter affinity than CHMP2A to mammalian Snf7 (CHMP4A) (*Effantin et al., 2013*).

To test these hypotheses, we first utilized MVB cargo sorting assays (for cargoes Mup1 and Can1) in *Saccharomyces cerevisiae* (*Banjade et al., 2019a*; *Banjade et al., 2019b*). In a *vps24Δ* strain overexpressing *VPS2*, we observed that Mup1-pHluorin is sorted at about 40% compared to that of the wild type (WT), and that the canavanine sensitivity of *vps24Δ* is partially rescued (*Figure 1A*). We also noted that *VPS2* overexpression rescues the temperature sensitivity of *vps24Δ* (*Figure 1—figure supplement 1A*), suggesting that the increased concentration of Vps2 could replace the cellular function of Vps24 beyond MVB cargo sorting.

With a Tet-off regulatable operator, we next used doxycycline to titrate the expression level of Vps2. As a result, we determined that about eightfold overexpression of Vps2 is necessary for restoring Mup1 sorting (*Figure 1—figure supplement 1B–D*). These data suggested that Vps2 contains features that can replace the function of Vps24 when present in higher concentrations in cells.

### Random mutagenesis and selection of *vps2* mutants that are capable of replacing both *VPS2* and *VPS24*

To identify the features in Vps2 that could replace the function of Vps24, we utilized an unbiased random mutagenesis selection approach, as we have done previously (*Banjade et al., 2019a*; *Tang et al., 2016*). Since *vps24Δ* is sensitive to the drug canavanine, we selected for *vps2* mutants that conferred canavanine resistance to *vps24Δ* cells. We performed error-prone PCR and assembled

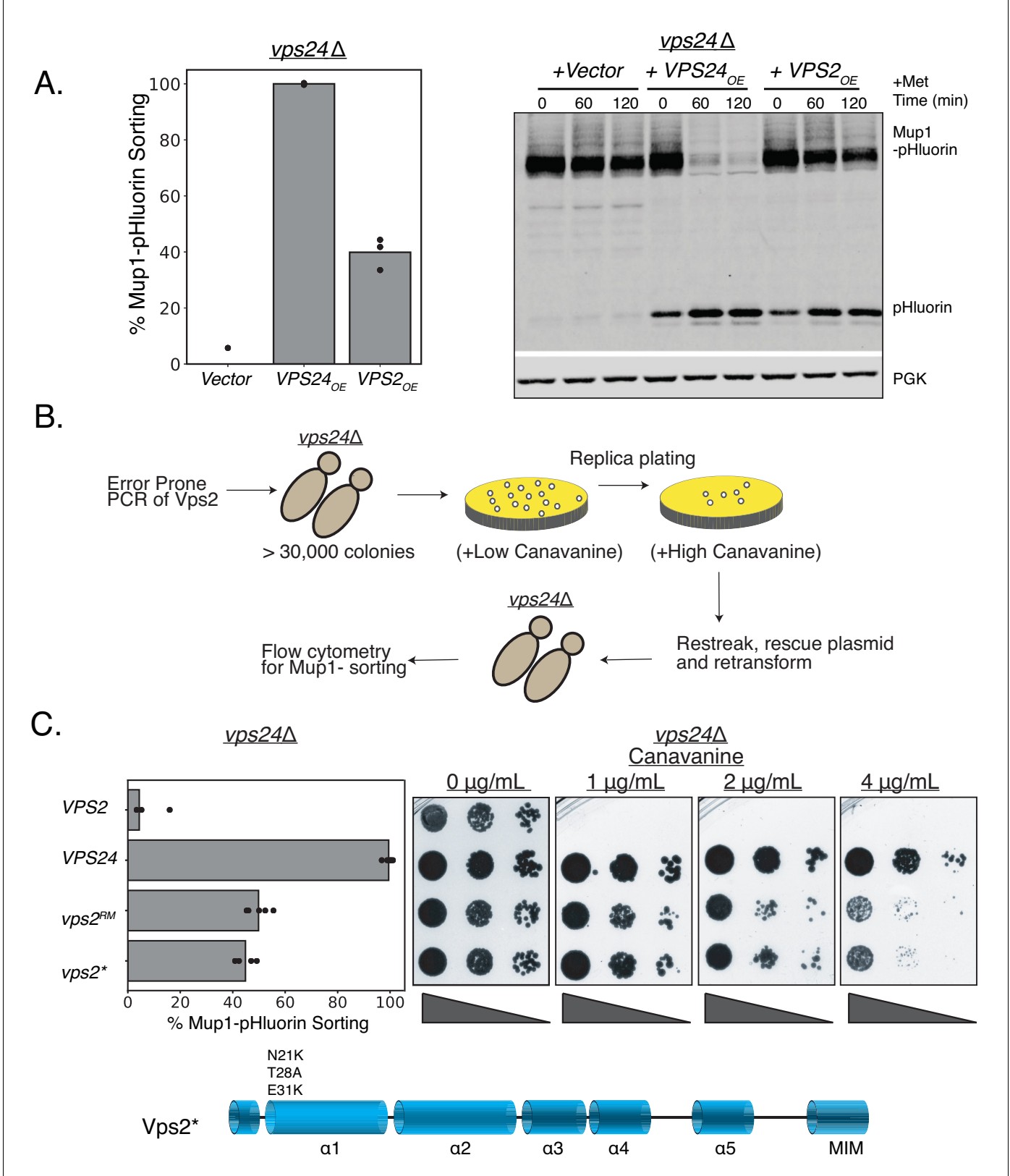

**Figure 1.** Minor modifications in Vps2 can replace the function of Vps24. (**A**) Overexpression of Vps2 can rescue the defect of *vps24Δ* for Mup1 sorting. Image on the left represents Mup-pHluorin sorting through a flow cytometry assay and the image on the right represents an immunoblot for pHluorin upon methionine addition. Overexpression (OE) was achieved through a CMV promoter and Tet operator containing plasmid. (**B**) Flowchart of the

*Figure 1 continued on next page*

*Figure 1 continued*

random mutagenesis approach. (C) Top figure shows the flow cytometry and canavanine sensitivity assays with the mutants of Vps2 that can rescue the sorting defects of *vps24Δ*. Bottom figure shows the domains of Vps2 highlighting the mutations in Vps2* .

The online version of this article includes the following source data and figure supplement(s) for figure 1:

**Source data 1.** Mup1-pHluorin sorting data associated with data in *Figure 1A*, *Figure 1C*, *Figure 1—figure supplement 1C*, *Figure 1—figure supplement 1D*, *Figure 1—figure supplement 3A*, *Figure 1—figure supplement 3B*.
**Figure supplement 1.** Overexpression of Vps2 can rescue the defect of *vps24Δ*.
**Figure supplement 2.** Helix-1 region of Vps2 is important for binding to Snf7.
**Figure supplement 3.** Vps2 N-terminal mutations can rescue the defect of *vps24Δ*.

a *vps2* mutant library in a *vps24Δ* strain. We next selected *vps2* alleles using canavanine at a concentration that the WT *VPS2* does not grow (*Figure 1B*). From this selection, one of the alleles (hereafter referred to as Vps2[RM]) strongly rescues the canavanine sensitivity of *vps24Δ* (*Figure 1C*) and sorts Mup1-pHluorin to 45% of that of the WT (*Figure 1C*). Vps2[RM] contains mutations in its promoter region, three missense mutations in helix 1 (N21K T28A E31K), and two missense mutations in helix 4 (S136N M146I).

The N-terminal mutations E21K T28A N31K (with the promoter mutations, hereafter called Vps2*, *Figure 1C* schematic) are necessary and sufficient for the suppression effect. Interestingly, these mutants also lie on the same surface of the alpha-1 helix, as a helical wheel representation suggests (*Figure 1—figure supplement 2B*). We found that while the individual mutations did not rescue *vps24Δ* (*Figure 1—figure supplement 3A*), they collectively suppressed both the defect in canavanine sensitivity and Mup1 (sorting up to ~40%). Because the mutant we isolated also had promoter mutations, the expression level of Vps2* is increased about threefold (*Figure 1—figure supplement 3C*). The suppression by Vps2* is not due simply to the overexpression effect however, since about eightfold overexpression of Vps2 is required for only ~20% sorting of Mup1 to occur in a *vps24Δ* strain (*Figure 1—figure supplement 1A–C*).

To test whether Vps2* possesses features of both Vps24 and Vps2, we performed cargo sorting assays in *vps24Δvps2Δ*. We observed that *vps2** also suppresses *vps24Δvps2Δ* (*Figure 2A–B*). Therefore, a synergistic effect of the mutations in the N-terminal basic region of Vps2 helix 1 and its threefold overexpression provides the necessary functional features of Vps24 and Vps2.

One of the early identified functions of Vps24 in yeast was as an adaptor for Vps2 to be recruited to Snf7 polymers (*Teis et al., 2008*). In the absence of Vps24, Vps2 does not bind to Snf7 (*Teis et al., 2008*; *Babst et al., 2002*). In co-immunoprecipitation experiments, Vps2* binds to Snf7 at a lower expression level than the WT Vps2, suggesting that the N-terminal helix-1 mutations increase the affinity of Vps2 for Snf7, bypassing the need for Vps24 (*Figure 2C-D*). Although the overall features of helix-1 region of Vps24 and Vps2 are similar (both basic helices), they vary in sequence composition (*Figure 1—figure supplement 2A*). Since the mutations occur in polar residues (*Figure 1—figure supplement 2A–B*), it is possible that these charge inversions increase the affinity of the basic patch of Vps2* to Snf7's acidic helix 4, consistent with our observations of Vps24 binding to Snf7 through an electrostatic interface (*Banjade et al., 2019a*). Simple overexpression of Vps2 probably rescues the defect of the lack of Vps24 since the overall concentration of a lower affinity Snf7 binding Vps2 molecule is increased in the cytoplasm.

Consistent with the model of helix-1 region of Vps24 and Vps2 interacting with helix-4 region of Snf7, Teis and colleagues report that defects resulting from charge inversion mutations in helix-4 region of Snf7 are rescued by helix-1 charge-inversion mutations in Vps2 (*Sprenger et al., 2021*). Collectively, these results provide strong evidence of the similarity between Vps24 and Vps2, supporting the idea that Vps2 associates with Snf7 in a lateral assembly mechanism, similar to Vps24.

## Binding to the AAA + ATPase Vps4 is a critical feature of the Vps24-Vps2 module

In contrast to Vps2 overexpression rescuing the defects of *vps24Δ*, the reverse does not occur – Vps24 overexpression by ~16-fold did not rescue the defect of a *vps2Δ* (*Figure 3*). One of the critical features of Vps2 is the presence of the C-terminal MIM motif that has a higher affinity to the AAA + ATPase Vps4 than other ESCRT-III proteins (*Obita et al., 2007*). While other ESCRT-III

proteins also possess the MIM motifs, the Vps2 MIM has the strongest affinity for Vps4 in solution (~20 µM, *Obita et al., 2007*). In addition, helix 5 of Vps2 has been identified as a second binding site of Vps4 with an affinity of ~3 µM (*Han et al., 2015*; *Han et al., 2017*; *Han et al., 2019*). We therefore hypothesized that a Vps24 variant with Vps4 binding sites could possess the properties of both Vps24 and Vps2. To test this, we replaced the C-terminus of Vps24 with that of Vps2 and assayed the functions of the chimeric protein.

We designed various chimeric constructs of Vps24/Vps2 under the control of their endogenous promoters to first demarcate the regions that maintain function in a *vps24Δ* strain (*Figure 3—figure supplement 1A*). Consistent with the sequence analysis, replacing the MIM and helix 5 of Vps24 with the homologous region of Vps2 kept the protein functional, but truncations beyond residue ~152 resulted in functional defects. In summary, the C-terminus (residues ~152 and beyond) of Vps24 can be replaced with that of Vps2 and still retain function.

We next tested these chimeras in *vps2Δ*. In contrast to *vps24Δ* cells, we observed that they do not functionally restore the loss of Vps2 (*Figure 3—figure supplement 1B*). To investigate whether they are dependent on protein expression levels, we then overexpressed these constructs with a CMV promoter that contain the N-terminus of Vps24 and helix 5 and MIM of Vps2. We found that they modestly suppress the defect of *vps2Δ* to about 30% of WT (*Figure 3*). Therefore, directly recruiting Vps4 on to Vps24 partially bypasses the requirement of Vps2. We speculated that there are additional features in Vps24 that make it distinct from Vps2.

ESCRT-III subunits are soluble monomers in the cytoplasm but undergo structural rearrangements when assembled on membranes. We next investigated whether activating mutations in Vps24 that induce conformational changes renders it similar to Vps2. In our previous work (*Tang et al., 2016*), we designed a Snf7 mutant (hereafter referred to as Snf7***) that rescues the defects of *vps20Δ*. Snf7*** includes a myristolyation site that recruits Snf7 to endosomes in the absence of upstream factors (ESCRT-I, ESCRT-II, and Vps20) (*Tang et al., 2016*), as well as missense mutations (R52E Q90L N100I) that trigger Snf7 to adopt an elongated, open, and membrane-bound conformation (*Henne et al., 2013*; *Buchkovich et al., 2013*).

Inspired by our early studies, we looked for single amino acid substitutions in Vps24 that weaken the interactions between helix 3 and helix 2 that would allow an extension of helices 2 and 3, and therefore 'activate' Vps24. As a result, we found that $vps24^{E114K}$ when overexpressed robustly rescues the defect of *vps2Δ* when it carries the Vps4 binding motifs of Vps2 (*Figure 3*). Taken together, our data suggest that a Vps24 variant capable of auto-activation and Vps4 recruitment can function as Vps2.

Our data that modifying Vps24 to include the Vps4 binding site and 'activating' mutations bypass Vps2 also led us to the natural hypothesis of whether we can bypass either or both Vps24 and Vps2 by including the Vps4 binding site directly onto Snf7 or on the activating mutant Snf7***. In other words, we asked whether the function of the Vps24-Vps2 module is simply to recruit Vps4 on to Snf7.

However, we found that these chimeras were unable to replace either Vps24 or Vps2 (*Figure 3— figure supplement 2*), suggesting that the function of Vps24-Vps2 goes beyond merely being adaptors to recruit the AAA + ATPase. In the Snf7-Vps2 chimeras we have made, helix 5 of Snf7 is replaced with that of Vps2, which is functional in a *snf7Δ* (*Figure 3—figure supplement 2*). Replacement of the MIM motif of Snf7 with that of Vps2 is also functional (*Figure 3—figure supplement 2*). Therefore, these replacements probably do not cause any structural destabilization of Snf7. However, replacing both the helix 5 and the MIM causes defects in Snf7 (Snf-Vps2H5MIM construct, *Figure 3—figure supplement 2*). This may happen because of an enhanced recruitment of Vps4, rapidly disassociating ESCRT-III polymers before vesicle scission can occur.

Therefore, the presence of a Vps24/Vps2-like molecule is critical for proper ESCRT-III function. Since previous publications (*Adell et al., 2017*; *Bertin et al., 2020*) and we ourselves (*Teis et al., 2008*) have found that Vps24-Vps2 act as laterally interacting proteins to Snf7 polymers that induce 3D helicity, an ESCRT-III functional protein requires this lateral copolymerization, perhaps to regulate Vps4-mediated subunit turnover of the heteropolymers.

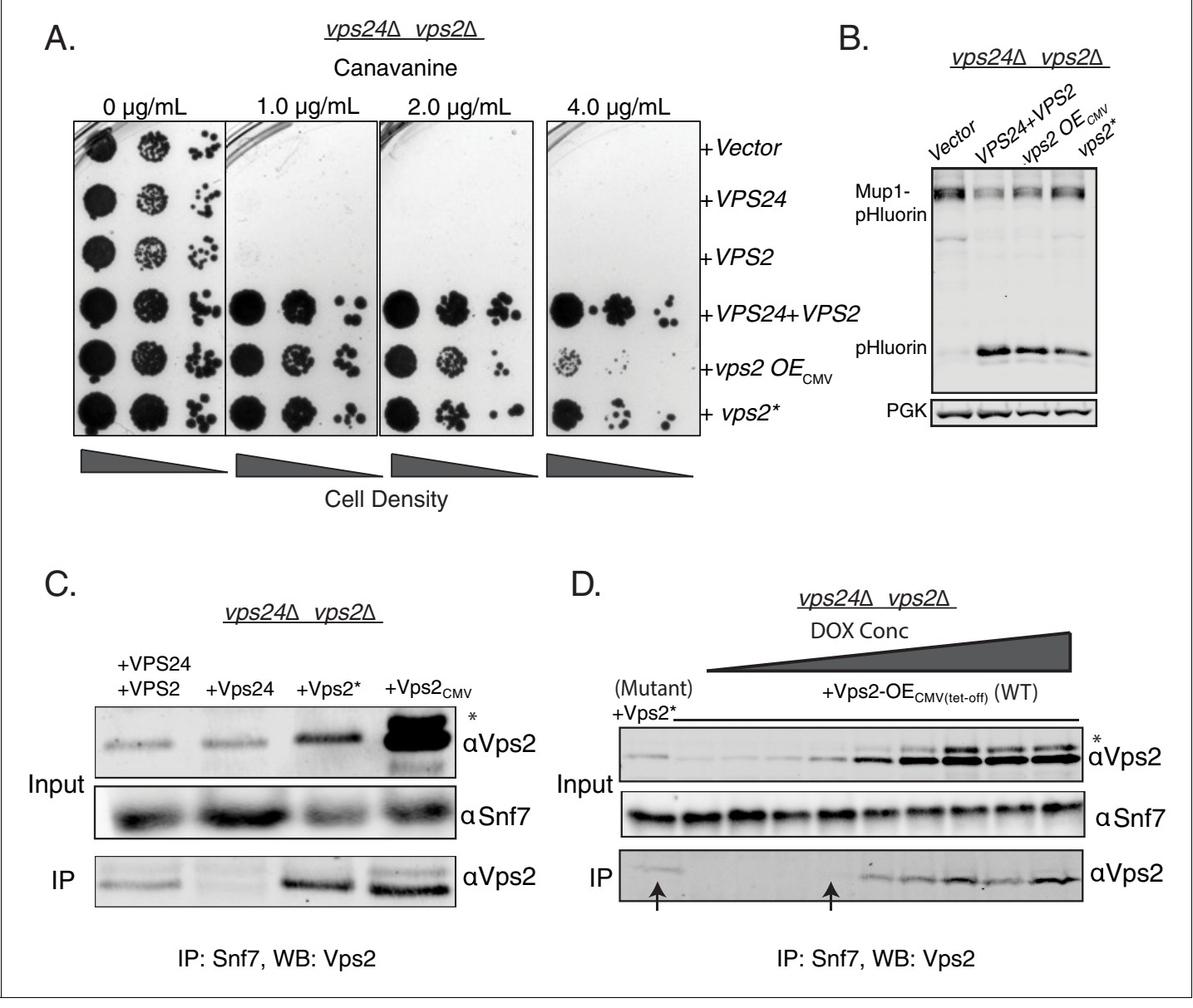

**Figure 2.** Properties of both Vps2 and Vps24 in a single Vps2 construct. (A) Canavanine sensitivity data in *vps24Δvps2Δ* with an overexpression of Vps2 (CMV-Tet system) or with Vps2*. (B) Immunoblot for Mup1-pHluorin sorting upon overexpression of Vps2 (CMV) or with Vps2*. (C) Co-immunoprecipitation of Snf7 with Vps2 (CMV) and Vps2* in *vps24Δvps2Δ*. (D) Co-immunoprecipation experiments of Snf7 with Vps2 at various expression levels of Vps2 after titration of the Tet-off operator with doxycycline. Arrows point to the relative binding to Snf at similar expression levels of Vps2 and Vps2*. In the gels, * refers to an unknown modified form of Vps2.

## Minor modifications in Vps24, most likely 'activating' mutations, can replace Vps2

In the 'closed' conformation of ESCRT-III subunits, the region from helix 3 and beyond bind back to alpha helices 1–2 (*Bajorek et al., 2009 McCullough et al., 2018*). It is hypothesized that during activation, the extension of helices 2–3 into an elongated helix triggers 'opening' of the protein, which enhances polymerization due to the availability of an extended surface for self-assembly. Therefore, mutations that trigger conformational changes to an 'open' state are able to enhance polymerization in vivo and in vitro. Consistent with these ideas, Vps24[E114K] in cell lysates (containing both the membrane and soluble fragments) forms higher molecular weight species in glycerol-gradient experiments (*Figure 4A*). In in vitro assays, while the WT Vps24 does not form polymers by itself or with

Vps2 (*Figure 4A*, *Figure 4—figure supplement 1C*), Vps24[E114K] readily associates into linear filaments with Vps2 (*Figure 4A*, *Figure 4—figure supplement 1A*). In comparison, previous experiments with Vps24[WT] required 70-fold higher concentrations to observe similar linear polymers (*Ghazi-Tabatabai et al., 2008*). With Snf7 and Vps2, both WT and the mutant Vps24 are able to form 3D spirals, which we previously described as the copolymeric structure of Snf7, Vps24, and Vps2 (*Figure 4—figure supplement 1B*). The increased ability to form polymers is consistent with the interpretation that E114K shifts the equilibrium of Vps24 to a polymerization-competent state. This polymerization-competent state, along with Vps4 binding sites, replaces the function of Vps2 in cells.

From sequence alignment analysis (*Figure 4C*), we noticed low conservation between Vps24 and Vps2 in the hinge between α2 and α3. Vps24 contains two potential helix-breaking asparagine residues in between these alpha helices (N99 and N103), while Vps2 lacks these helix-breaking residues (*Figure 4C*). We found that mutating the Asn to helix-stabilizing Ala (in addition to the Vps4 binding motifs) in Vps24 rescues the defect of *vps2Δ*, while the helix-breaking glycine does not rescue the defect (*Figure 4B*). Therefore, it appears that the hinge region, that contains residues which may affect the conformational flexibility of ESCRT-III proteins, in addition to the Vps4 binding sites, accounts for the majority of the difference between Vps24 and Vps2. We note that these data are

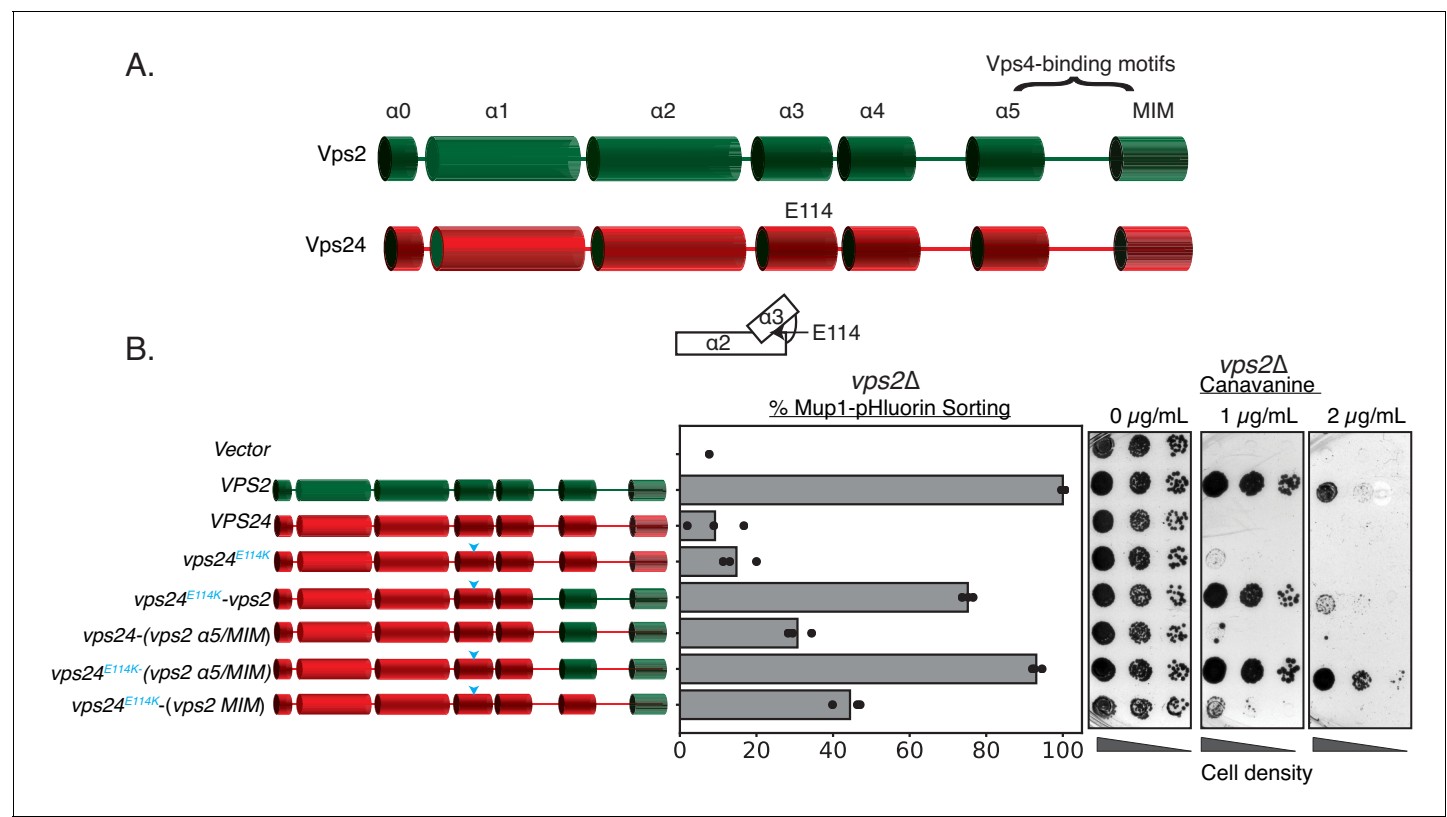

**Figure 3.** Simple modifications in Vps24 can be made to mimic Vps2. (**A**) The domain organization of Vps2, highlighting the C-terminal region important for Vps4 binding. (**B**) Left panel denotes the chimeras made to replace regions of Vps2 onto Vps24. Cyan arrows in the helices are positions of the E114K mutation. Right panel represents Mup1-pHluorin sorting and canavanine sensitivity assays. In this assay, the constructs were overexpressed under a CMV promoter-Tet-off operator system.

The online version of this article includes the following source data and figure supplement(s) for figure 3:

Source data 1. Mup1-pHluorin sorting data associated with figure in *Figure 3B*, *Figure 3—figure supplement 1A*, *Figure 3—figure supplement 1B*, *Figure 3—figure supplement 3A*, *Figure 3—figure supplement 3B*.
Figure supplement 1. Chimeras of Vps24-Vps2 are functional proteins.
Figure supplement 2. Simply adding Vps4 binding sites to Snf7 do not replace the functions of Vps24-Vps2.
Figure supplement 3. Simple modifications in Vps24 can be made to mimic Vps2.

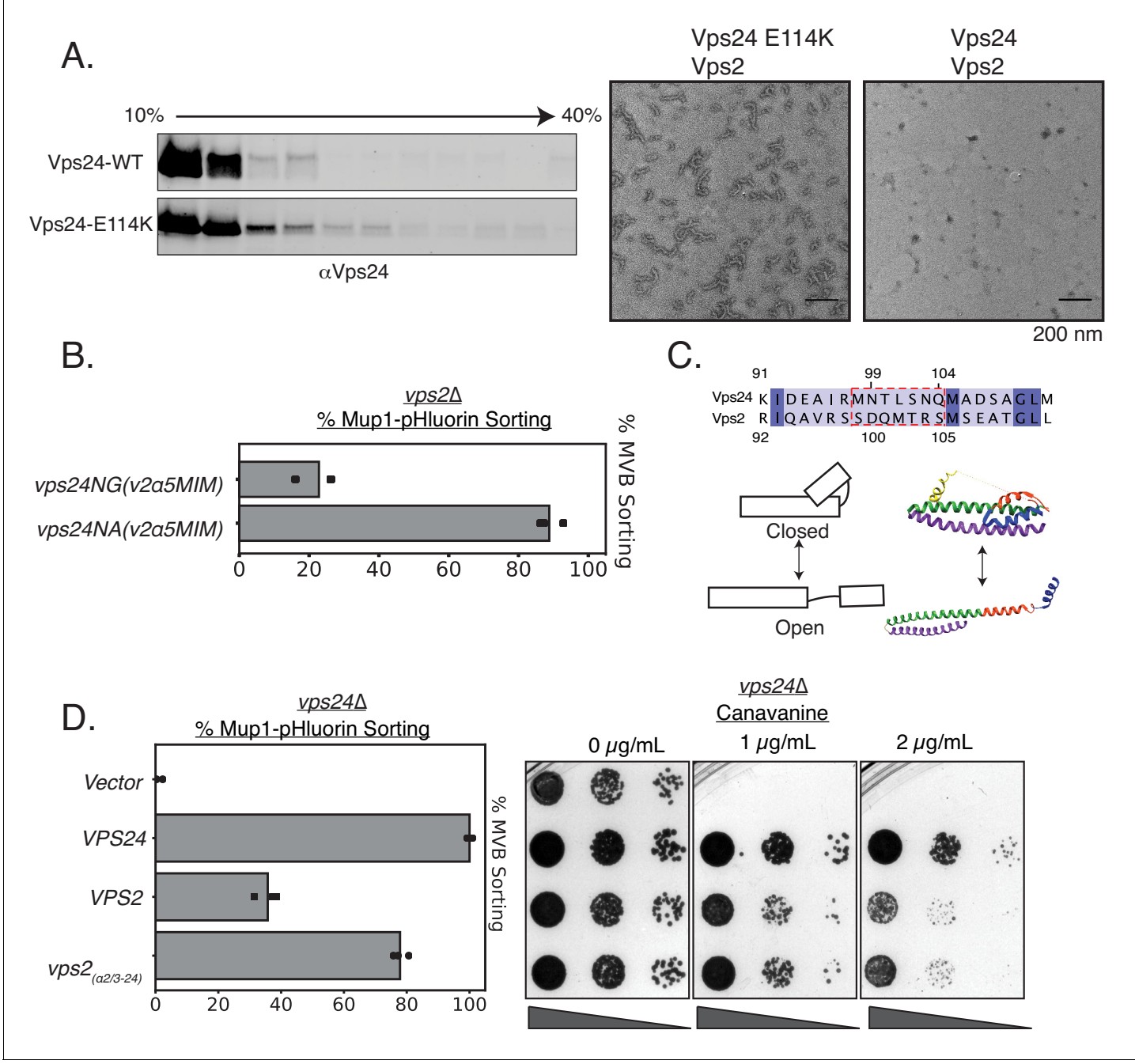

**Figure 4.** Vps24 and Vps2 may exhibit different conformations. (**A**) Left: Glycerol-gradient experiments with Vps24 and Vps24E114K suggest that the mutant can form higher molecular-weight species. Right: Negative stain electron microscopy of Vps24 E114K or wild-type (WT) Vps24 at 1 μM each of the proteins in the presence of Vps2. (**B**) Mup1-pHluorin assays with Vps24 mutations in the asparagines (N99 and N103) α2/α3 hinge region to Ala or Gly residues in constructs that have the Vps4 binding sites H5 (helix 5) and MIM from Vps2 (V2). These constructs are expressed with the CMV promoter, Tetoff system. See **Figure 3** for direct comparison with other Vps24 mutants and chimeras. (**C**) Top: Sequences of the α2/α3 hinge region of Vps24 and Vps2. Bottom left: Model showing the two conformations of ESCRT-III proteins. Structural model on the right is that of CHMP3 (closed) (**Bajorek et al., 2009**) and of Snf7 (open) (**Tang et al., 2015**). (**D**) Mup1-pHluorin sorting and canavanine sensitivity assays with overexpression of Vps2 (CMV-Tet) and with a mutant replacing the α2/α3 hinge region of Vps2 with that of Vps24 (also CMV-Tet system).

The online version of this article includes the following source data and figure supplement(s) for figure 4:

**Source data 1.** Mup1-pHluorin sorting data associated with figure in **Figure 4B** and **Figure 4C**.

**Figure supplement 1.** Vps24 (E114K) associates with Vps2.

**Figure supplement 2.** Structures of the autoinhibited CHMP3, and the filament forming conformations of Vps24 and Snf7.

indicative of mutations that control the conformations of the proteins. However, further biophysical analyses will be required for definitive evidence of this conformational flexibility.

Consistent with this, when we overexpress a variant that replaces the hinge region of Vps2 with that of Vps24, it suppresses *vps24Δ* to ~80% as tested by Mup1 sorting, compared to ~40% for that of the WT Vps2 (*Figure 4D*).

Collectively, these data suggest that Vps24 and Vps2 contain similar features. Laterally interacting with Snf7, inducing the formation of an ESCRT-III super-helix, and recruiting Vps4 are three features for the Vps24-Vps2 module.

## 'Accessory' ESCRT-III genes promote intraluminal vesicle formation

In *S. cerevisiae*, there are eight ESCRT-III genes. One of the defining features of these ESCRT-III proteins is the N-terminal alpha-helical bundle, which is sometimes referred to as the ESCRT-III domain. The other defining feature is the C-terminal flexible region that contains at least one MIM, which binds to the MIT domain of Vps4. The N-terminal regions, the ESCRT-III domains, are similar in sequence and structure in the eight ESCRT-III proteins. However, the specific functions of all these ESCRT-III proteins remain unclear.

To quantitatively assess the relative contributions among the ESCRT proteins, we assayed for Mup1-pHluorin sorting in each gene deletion (*Figure 5—figure supplement 1*). We also performed canavanine sensitivity assays (*Figure 5—figure supplement 1B*) with the same mutants. We observed that *snf7Δ*, *vps20Δ*, *vps24Δ*, and *vps2Δ* show severe sorting defects, *did2Δ* and *vps60Δ* show partial sorting defects, and *ist1Δ* and *chm7Δ* show no defect. These data are consistent with previous findings with a different cargo (CPS) (*Rue et al., 2008*), and from in vitro assays (*Schöneberg et al., 2018*; *Pfitzner et al., 2020*), which suggest that Snf7, Vps20, Vps24, and Vps2 are the minimal contributors in MVB formation.

These differences in function during MVB formation occur despite similarity in structure and sequence between these ESCRT-III proteins. Inspired by the observation that *VPS2* overexpression suppresses *vps24Δ*, we investigated whether overexpressing other ESCRT-III genes could suppress the deletions of a different ESCRT-III gene. We used the CMV promoter to overexpress each of the ESCRT-III genes: Snf7 is overexpressed by ~5-fold, and Vps24 and Vps2 by ~16-fold. Most of the overexpression constructs did not rescue the defect of the other ESCRT-III deletions, except in two cases (*Figure 5*). As described above, Vps2 overexpression rescued the defect of *vps24Δ* and also partially rescued the defect of a *did2Δ* (*Figure 5*).

Evolutionary analyses have grouped ESCRT-III into two groups: Snf7-Vps20-Vps60 and Vps24-Vps2-Did2 (*Figure 6A*; *Leung et al., 2008*; *Caspi and Dekker, 2018*). Vps20 nucleates formation of Snf7 spirals, and Vps24-Vps2 induces bundling and helix formation of spirals (*Banjade et al., 2019a*). In in vitro assays with lipid monolayers, we found that Did2 forms tube-like helices (*Figure 6B*, and as previously shown in *McCullough et al., 2015*; *Nguyen et al., 2020* for mammalian Did2 named CHMP1). However, Vps60 lacks the ability to form long helices/tubes, and preferentially forms spiral-like structures, reminiscent of Snf7 (*Figure 6B*). Consistent with Vps60 mimicking Snf7 structurally, the N-terminal region of Snf7 fused to the C-terminal region of Vps60 rescues the defects of *vps60Δ* (*Figure 6C*). Vps60-GFP is localized to endosomal and vacuolar membranes with a hint of plasma membrane signal (*Figure 6—figure supplement 1*). This localization is primarily cytosolic in *vps20Δ*, *snf7Δ*, or *vps2Δ*, and unchanged in *did2Δ* (*Figure 6—figure supplement 1*).

In summary, among Vps20-Snf7-Vps20, Snf7 serves as the main scaffold which can be engineered to substitute for Vps20 (*Tang et al., 2016*) or Vps60 (*Figure 6A*). Among Vps24-Vps2-Did2, modifications within Vps24-Vps2 can functionally replace each other; Did2 resembles Vps24-Vps2 as it readily forms 3D helices, and *did2Δ* can be partially suppressed by Vps2 overexpression. Our data suggest that although ESCRT-III subunits have evolved for divergent roles in ordered assembly, rational modifications in ESCRT-III subunits can allow one to consolidate the functions of two ESCRT-III proteins into one ESCRT-III protein.

## Discussion

In this work we have utilized rational design and unbiased mutagenesis to understand the design principles of ESCRT-III subunits. To simplify this larger question, we focused primarily on the ESCRT-III subunits Vps24 and Vps2. First, we find that overexpression of Vps2, by approximately eight fold

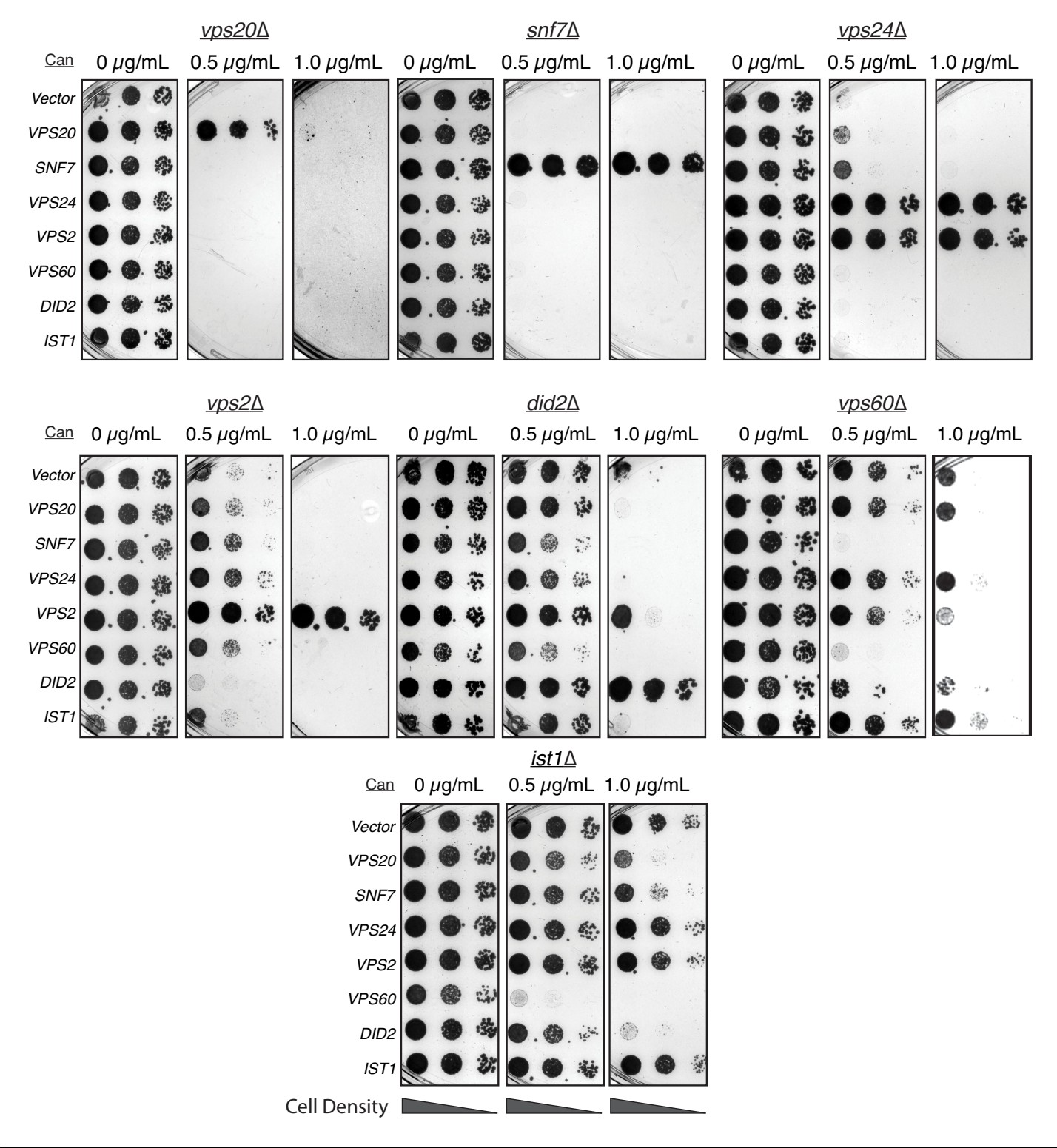

**Figure 5.** Overexpressing ESCRT-III proteins in the background of other ESCRT-III mutants show selective rescue phenotypes. In the annotated mutants, ESCRT-III proteins were expressed with a CMV promoter/Tet operator system and plated in canavanine-containing plates. Vps2 overexpression can rescue the defect of *vps24Δ*. Vps2 overexpression in a *did2Δ* partially rescues canavanine sensitivity. Vps60 overexpression appears to be dominant negative.

The online version of this article includes the following source data and figure supplement(s) for figure 5:

*Figure 5 continued on next page*

*Figure 5 continued*

**Figure supplement 1.** Relative effects of all ESCRT-III mutants for defects in cargo sorting.

**Figure supplement 1—source data 1.** Mup1-pHluorin sorting data associated with figure in *Figure 5—figure supplement 1A*.

and above, can rescue *vps24Δ* in yeast. Second, point mutations in the helix-1 region of Vps2 can also rescue *vps24Δ*. These Vps2 mutants also bind to Snf7 in vivo even in the absence of Vps24. Third, overexpression and inclusion of higher affinity AAA + ATPase Vps4 binding regions on Vps24 can rescue the absence of Vps2. Fourth, mutations that 'activate' Vps24-Vps2, by possibly inducing conformational changes, also rescue each other's function. These data indicate a strong similarity in-between these two ESCRT-III subunits.

Our data suggest that Vps24 may not possess the ability to fully extend its helices 2 and 3 in the ESCRT-III copolymer. A recently published CryoEM structure showed that the Vps24 homopolymer consists of Vps24 protomers in a 'semi-open' conformation (*Huber et al., 2020*) (see *Figure 4—figure supplement 2* for direct comparison), in contrast to the fully extended and open Snf7 (*Tang et al., 2015*) and Did2 (CHMP1) (*Nguyen et al., 2020*) polymers. It is possible that a mixture of different conformations allows for efficient Vps24-Vps2 assembly, which has a higher affinity to the Snf7 polymer.

Despite the observed similarity between Vps24 and Vps2, our data also suggest some differences: they may exist in different conformations, and that Vps2 consists of a higher affinity Vps4 binding site, consistent with previous mammalian CHMP2 and CHMP3 work (*Effantin et al., 2013*). In addition, Vps24-Vps2 induce lateral association and bundling (*Mierzwa et al., 2017*; *Pfitzner et al., 2020*; *Banjade et al., 2019a*), along with helicity of the spirals, which could be an important parameter for ESCRT-III function for MVB biogenesis.

Heteropolymerization of Vps2 and Vps24 into an elongated polymer, interacting 'longitudinally' with the helices 2 and 3, is also consistent with the model presented in the manuscript by Teis and colleagues (*Sprenger et al., 2021*). In their work, the authors show through crosslinking experiments that site-specific cysteine mutations in helices 2 and 3 of Vps24 and Vps2 crosslinked with one another, and charge inversion mutations in helix 2/3 also rescued each other's defects. Therefore, similar to Snf7's 'activating' mutations triggering its longitudinal polymerization through helix 2/3 (3, 8, 29), Vps24's 'opening' could also trigger longitudinal heteropolymerization with Vps2. We note that this model requires further corroboration with structural and biophysical analyses.

Following these observations, we propose the following three critical aspects of an ESCRT-III minimal module: (1) a core spiral forming unit (e.g., Snf7), (2) a lateral bundling unit (e.g., Vps24 and Vps2), and (3) an ability to recruit a disassembly machinery (e.g., the AAA + ATPase Vps4) (*Figure 7B*).

In addition to the rescue of function phenotypes with Vps24-Vps2, we also find that *did2Δ* can be rescued partially with the overexpression of Vps2. Similarly, swapping the C-termini of Snf7 with that of Vps60 can replace the function of *vps60Δ*. We previously showed that point mutations in Snf7 can rescue the absence of Vps20 (*Tang et al., 2016*). These data imply, as predicted by evolutionary analyses (*Leung et al., 2008*; *Spang et al., 2015*), that Vps20-Snf7-Vps60 are more similar to one another, and Vps2-Vps24-Did2 are more alike one another, given the rescue of *vps20Δ* or *vps60* with Snf7 alleles, and *vps24Δ* and *did2Δ* with Vps2 alleles.

In vitro reconstitutions suggest that Snf7, Vps24, and Vps2 are essential for membrane budding, and Vps4 for vesicle scission (*Schöneberg et al., 2018*; *Pfitzner et al., 2020*). Vps4-mediated turnover of a laterally associating and helix-inducing polymer of Vps24-Vps2 would constrict the Snf7 scaffold to a fission-competent structure, as predicted by simulations (*Harker-Kirschneck et al., 2019*) and as recently proposed to occur in archaeal cell division (*Tarrason Risa et al., 2020*). While Did2 and Ist1 are not essential for intraluminal vesicle formation in vitro and in vivo, they have regulatory roles that are controlled by sequential polymerization dynamics through Vps4 (*Pfitzner et al., 2020*). So far Vps60 has not been included in in vitro analyses and there have been fewer in vivo analyses on this protein. We find that for Vps60 recruitment to endosomal/vacuole membranes, Vps2 (and likely Vps24) is required. Therefore, Vps60 may be recruited in later stages of polymer formation, in a sequential fashion to the core scaffold, as Did2 does (*Pfitzner et al., 2020*). Vps60 has been shown to interact with Vta1 (*Yang et al., 2012*; *Azmi et al., 2008*; *Nickerson et al., 2010*),

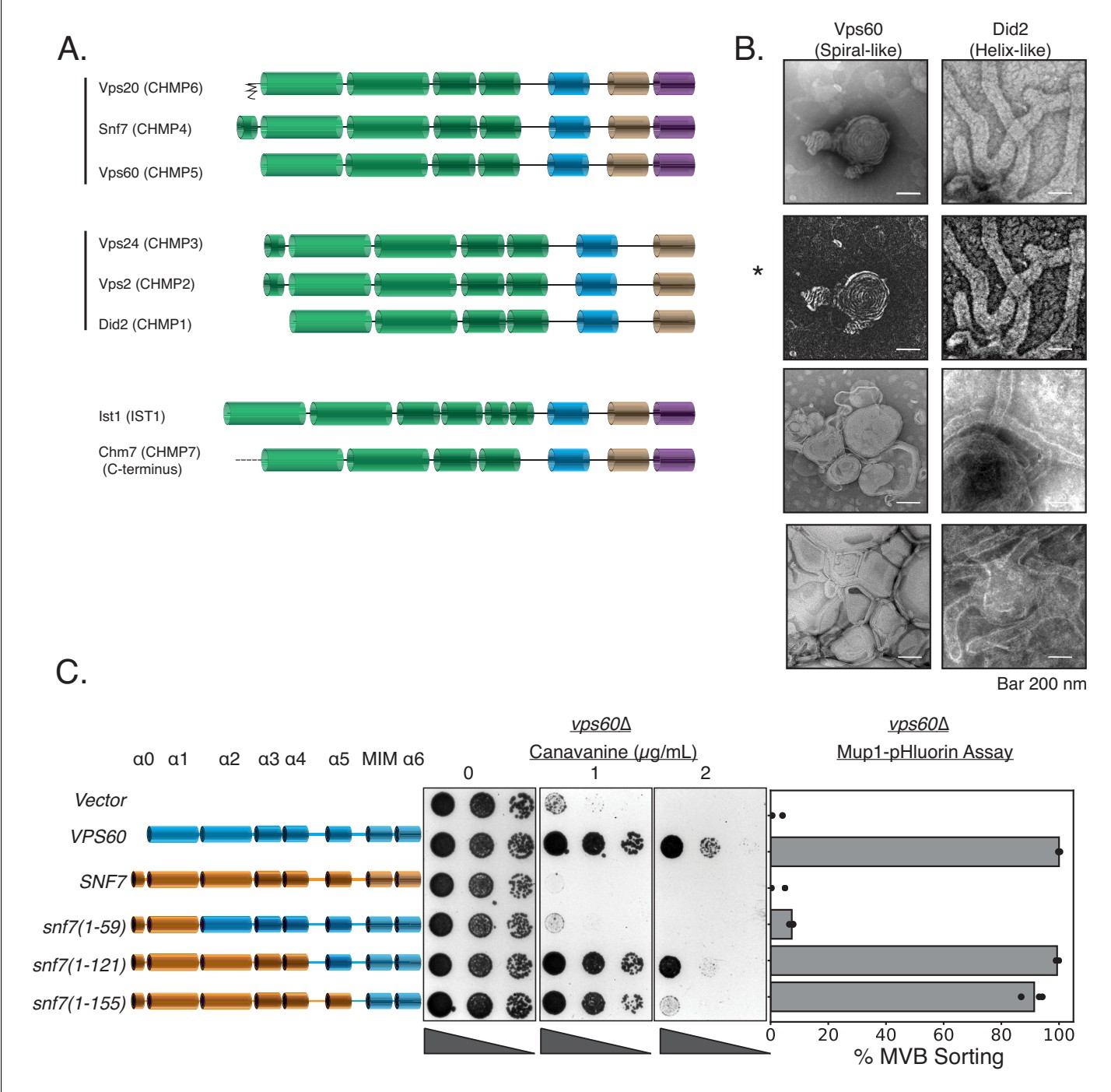

**Figure 6.** Vps60 possesses features of Snf7. (**A**) Domain subunits of the eight ESCRT-III proteins in yeast. The mammalian names are in parentheses. (**B**) Electron microscopy images of 1 μM Vps60 or 1 μM Did2 on lipid monolayers, incubated for 1 hr. Bar is 200 nm each. Top two images (highlighted by an asterisk, *) are different contrast-adjusted depictions of the same image. (**C**) Domain swaps from Snf7 onto Vps60 can rescue the defects of canavanine sensitivity and Mup1-pHluorin sorting in a vps60Δ strain.

The online version of this article includes the following source data and figure supplement(s) for figure 6:

**Source data 1.** Zip file contains Mup1-pHluorin sorting data associated with figure in *Figure 6—figure supplement 1C* and the raw images used to make *Figure 6—figure supplement 1*.

**Figure supplement 1.** Localization of *VPS60-GFP* in different ESCRT-III mutants.

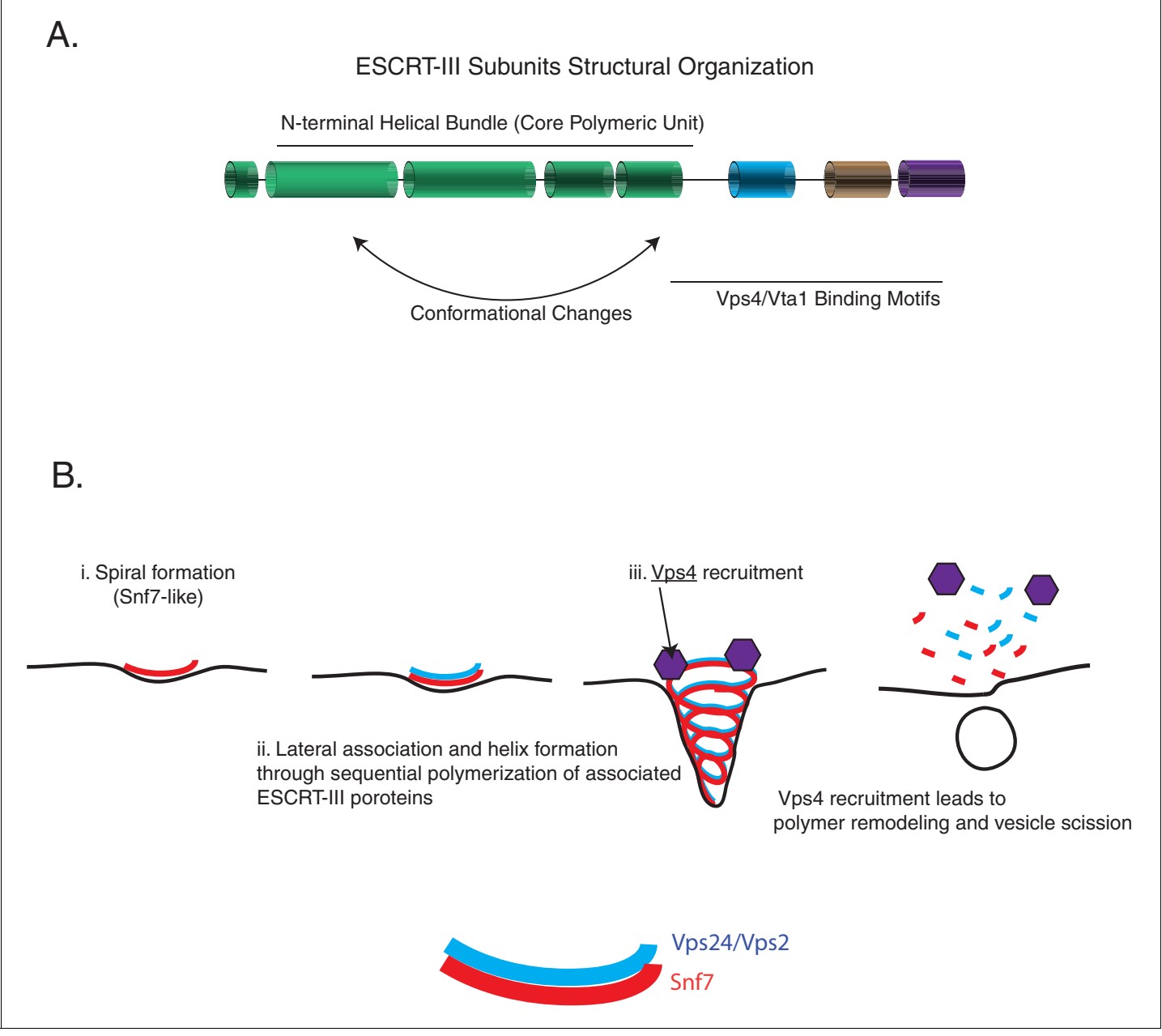

**Figure 7.** ESCRT-III assembly principles. (**A**) The domain organization of ESCRT-III subunits and the various functional parts of the structures/sequence. (**B**) The minimal features of ESCRT-III assembly may involve spiral formation, lateral association between copolymers that induce helicity, and recruitment of a disassembly factor such as Vps4. Schematic on the bottom shows how spirals of Snf7 and Vps24/Vps2 may laterally associate.

and Vta1 is known to be an activator of Vps4 (*Yang et al., 2012*; *Norgan et al., 2013*; *Monroe et al., 2017*). Therefore, it is possible that Vps60 is involved in further activating Vps4 function in the later stages of polymer dynamics.

ESCRT-III proteins are integral to all ESCRT-related functions in cells. However, the mechanisms of the specific roles of each ESCRT-III proteins have remained unclear. Based on our work on Vps24 and Vps2 described here and our previous studies on Snf7 and Vps20 (*Tang et al., 2016*), the molecular features of ESCRT-III subunits should enable future work on rational design of minimal ESCRT-III subunit(s) possessing all properties necessary for intraluminal vesicle formation. Further in vivo analyses and in vitro reconstitution are required to test whether this minimal ESCRT-III subunit(s) can be created that include the aforementioned features. Some archaeal species consist of only two ESCRT-

III proteins, which must possess the minimal properties of ESCRT-III necessary for function (*Caspi and Dekker, 2018*). With the principles learned from our work and from recent studies on ESCRT-III, it will be interesting to study what biochemical features the ancient archaeal ESCRT-III subunits contain, and what additional features were acquired as eukaryotic organelles evolved.

Our earlier understanding of Vps24 and Vps2 suggested that they bound cooperatively to Snf7, but that these were independent proteins with independent and specific functions for MVB sorting. Our data in this study suggest that minor modifications to either one can replace the function of another. These data provide an explanation for why in certain biological processes CHMP3 (mammalian Vps24) may play a minor role (such as in HIV budding), as the isoforms of CHMP2 (Vps2) may already possess the ability to form lateral interactions, and also an ability to recruit the AAA + ATPase Vps4. The relative contributions of the different ESCRT-III proteins for other ESCRT-dependent processes have not been quantified to the same extent. Further analysis of site-specific ESCRT-III function could allow us to achieve targeted cellular manipulation of ESCRT-dependent processes, understanding of the evolution of these membrane remodeling polymers and how they contribute to organelle biogenesis.

## Materials and methods

### Random mutagenesis selection
Error-prone PCR was used to generate random mutations in the plasmid harboring the *S. cerevisiae* Vps2 gene. The primers used for this PCR bind the 5' UTR and the 3' UTR regions of Vps2. The PCR fragment was transformed into the *vps24Δ* strain in the presence of a linearized Vps2 plasmid by digesting with HindIII and NarI enzymes. The transformants were plated onto plates containing 0.5 µg/mL canavanine, and then replica plated into 4 µg/mL plates. Plasmids were rescued from these colonies that grew on canavanine and then re-transformed into the *vps24Δ* strain to confirm the suppression of *vps24Δ*. Confirmation of *vps24Δ* suppression was done by testing Mup1-pHluorin sorting ability (see below) of the mutants.

### Canavanine spot plates
Mid-log cells were serially diluted to an $OD_{600}$ of 0.1. They were then diluted 10-fold serially, and spot-plated in plates containing various concentrations of canavanine. Images of the plates were taken at 3 and 5 days.

### Mup1-pHluorin flow cytometry and immunoblots
Strains harboring Mup1-pHluorin were used to assay endocytosis of this cargo upon methionine addition. Assays were performed as described before (*Tang et al., 2016*). Briefly, mid-log cells in the presence of synthetic drop-out media were treated with 20 µg/mL L-methionine for 90 min and assayed for quenching of pHluorin. Over time as Mup1-pHluorin traffics to the vacuole, fluorescence decreases due to quenching of the pH-sensitive pHluorin. Experiments were performed at room temperature and analyses were done on a C6 Accuri flow cytometer from BD Biosciences; 100,000 cells were used for each measurement, and the mean fluorescence was used for analyses. Mup1-pHluorin sorting data (*Figure 1A and C*, *2B*, *Figure 1—figure supplements 1C*, *2B*, *Figure 3—figure supplements 1A*, *Figure 3—figure supplements 1B*, *Figure 3—figure supplement 3A*, *Figure 3—figure supplement 3B*, *Figure 4B and D*, *Figure 6C*) presented contains bar graphs representing averages, and also the independent replicate denoted by scatter plots. Error bars, when present, in *Figure 1—figure supplement 3*, and *Figure 5—figure supplement 1A* represent standard deviation from three independent experiments.

Immunoblots after methionine treatment were performed to analyze free pHlourin, upon degradation of Mup1, as described (*Banjade et al., 2019a*). Blots were performed using primary antibody against GFP from Torrey Pines. Imaging of the western blots was performed using an Odyssey CLx imaging system and analyzed using the Image Studio Lite 4.0.21 software (LI-COR Biosciences).

### Doxycycline-mediated shutdown of the Tet-off operator
Plasmids (pCM189) used in this study that under the Tet-off operator have a CMV promoter and can be regulated by doxycycline titration. For titration experiments, cells were diluted to an $OD_{600}$ of

0.01. Doxycyline was added at a concentration of 0.25 µg/mL and serially diluted twofold over eight times. Cultures were grown until an $OD_{600}$ of 0.5, and then treated with methionine for Mup1-pHlourin sorting assays or used for co-immunoprecipitation. Error bar for quantitation of overexpression through doxycycline titration in *Figure 1—figure supplement 1D* represents standard deviation from three independent blots.

### Protein purification

Vps24, Snf7[R52E], and Vps2 constructs used in this study were purified as described before (*Banjade et al., 2019a*). A combination of affinity (Cobalt Talon resin) and size exclusion chromatography (SD200increase, GE) were used to purify the proteins. The final buffer under which the proteins are stored was 25 mM Hepes pH 7.5, 150 mM NaCl, and 2 mM β-ME. His6-tagged Vps60 and Did2 were purified through cobalt and anion exchange chromatography.

### Electron microscopy

Lipid monolayers were prepared with a mixture of 60% POPC, 30% POPS, and 10% PI3P in chloroform. Carbon-coated electron microscope grids were used to make monolayers and incubate with proteins, as described before (*Banjade et al., 2019b*). Grids were stained with 2% ammonium molybdate and imaged on an FEI Morgagni 268 TEM.

### Glycerol gradient

For Vps24 and Vps24 E114K glycerol gradients, *vps24Δ* was transformed with pCM189 Vps24 or pCM189 Vps24 E114K. Thirty ODs of cells expressing these constructs were harvested in phosphate saline buffer (PBS). Lysis was performed with PBS buffer, 10% glycerol, 1 mM DTT, Roche protease cocktail, and 0.5% Tween-20. Gradient Master 108 from Biocomp was used to make glycerol gradients of 10–40%. Centrifugation was performed at 100,000 × *g* for 4 hr at 4°C; 1 mL fractions were collected from the solutions, TCA-precipitated, and immunoblotted.

### Fluorescence microscopy

One milliliter of mid-log cells was harvested and resuspended in 25 µL water. Imaging was performed on a Deltavision Elite system with an Olympus IX-71 inverted microscope, using a 100×/1.4 NA oil objective. Image extraction and analysis were performed using the FiJi software.

### Sequence and structural analyses

Mafft (*Katoh et al., 2002*) and Jalview (*Clamp et al., 2004*) were used to analyze sequences. Heliquest was used for helical wheel analysis (*Gautier et al., 2008*). Structural models were made using UCSF Chimera (*Pettersen et al., 2004*).

### Co-immuoprecipitation

Thirty ODs of mid-log cells were harvested and washed with cold MilliQ $H_2O$, and resuspended in 1 mL PBS, 10% glycerol, 1 mM DTT, and 1 mM EDTA, including protease inhibitor cocktail from Roche. Lysis was performed by bead-beating (Zirconia-Silicon beads) twice for 30 s, with 30 s intervals on ice. Lysate was treated with 1% Triton X-100 and rotated for 20 min. Lysate was cleared by centrifugation at 16,000 × *g* at 4°C. The supernatant was treated with protein G beads (Dynabeads) for 30 min at 4°C to remove nonspecific binding. The magnetic beads used for this assay were allowed to settle with a magnetic Eppendorf-tube rack, and the supernatant was applied with 1/250 v/v of anti-Snf7 antibody. After 1 hr incubation at 4°C, the beads were washed twice with 20×-fold bead volume of the lysis buffer. Proteins were eluted by incubating the beads at 65°C for 10 min in sample buffer (150 mM Tris-Cl, pH 6.8, 8 M urea, 10% SDS, 24% glycerol, 10% v/v β-ME, and bromophenol blue). Anti-Snf7 and anti-Vps2 antibodies were used to probe for eluted proteins through Western blots.

## Acknowledgements

We thank David Teis for the gift of anti-Vps2 antibody and for sharing unpublished data. We thank all members of the Emr lab for discussions. Work in the Emr lab is supported by a Cornell University

Research Grant CU3704. Sudeep Banjade is an HHMI fellow of the Damon Runyon Cancer Research Foundation (DRG-2273–16). We are also grateful to the Damon Runyon Cancer Research Foundation for an extension of the fellowship to support our work during the COVID-19 pandemic delays. Shaogeng Tang is a Merck fellow of the Damon Runyon Cancer Research Foundation (DRG-2301–17) on a different project.

## Additional information

### Funding

| Funder | Grant reference number | Author |
|---|---|---|
| Damon Runyon Cancer Research Foundation | DRG-2273-16 | Sudeep Banjade |
| Cornell University | CU3704 | Scott D Emr |

The funders had no role in study design, data collection and interpretation, or the decision to submit the work for publication.

### Author contributions

Sudeep Banjade, Conceptualization, Data curation, Formal analysis, Funding acquisition, Validation, Investigation, Methodology, Writing - original draft, Writing - review and editing; Yousuf H Shah, Formal analysis, Investigation; Shaogeng Tang, Conceptualization, Formal analysis, Writing - review and editing; Scott D Emr, Conceptualization, Supervision, Funding acquisition, Writing - review and editing

### Author ORCIDs

Sudeep Banjade (iD) https://orcid.org/0000-0002-5920-891X
Shaogeng Tang (iD) http://orcid.org/0000-0002-3904-492X
Scott D Emr (iD) https://orcid.org/0000-0002-5408-6781

### Decision letter and Author response

Decision letter https://doi.org/10.7554/eLife.67709.sa1
Author response https://doi.org/10.7554/eLife.67709.sa2

## Additional files

### Supplementary files

• Transparent reporting form

### Data availability

All data generated are included in the study.

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

# Appendix 1

**Appendix 1—key resources table**

| Reagent type (species) or resource | Designation | Source or reference | Identifiers | Additional information |
|---|---|---|---|---|
| Strain, strain background (*Saccharomyces cerevisiae, Matα*) | WT | PMID:3062374 | SEY6210 | (*Background strain*) *MATα ura3-52 his3-200 leu2-3,112 trp1-901 lys2-801 suc2-9* |
| Strain, strain background (*S. cerevisiae, Mata*) | WT | PMID:3062374 | SEY6210.1 | (*Background strain*) *MATa leu2-3,112 ura3-52 his3-Δ200 trp1-Δ901 lys2-801 suc2-Δ9* |
| Strain, strain background (*S. cerevisiae, Mata*) | WT; *Mup-pHluorin* | PMID:24139821 | NBY40 | (*SEY6210.1*); *MUP1-pHLUORIN::KANMX* |
| Strain, strain background (*S. cerevisiae, Mata*) | *snf7Δ* | PMID:23063125 | NBY44 | (*SEY6210.1*); *snf7Δ::HIS3; MUP1-PHLUORIN::KAN* |
| Strain, strain background (*S. cerevisiae, Mata*) | *vps24Δ* | PMID:24139821 | NBY47 | (*SEY6210.1*); *vps24Δ::HIS3; MUP1-pHLUORIN::KANMX* |
| Strain, strain background (*S. cerevisiae, Mata*) | *vps2Δ* | PMID:24139821 | NBY69 | (*SEY6210.1*); *vps2Δ::HIS3 MUP1-pHLUORIN::KANMX* |
| Strain, strain background (*S. cerevisiae, Mata*) | *did2Δ* | This study | STY35 | (*SEY6210.1*); *did2Δ::TRP1; Mup1-pHluorin::KAN* |
| Strain, strain background (*S. cerevisiae, Mata*) | *chm7Δ* | This study | STY25 | (*SEY6210.1*); *yjl049wΔ::TRP1; Mup1-pHluorin::KAN* |
| Strain, strain background (*S. cerevisiae, Matα*) | *ist1Δ* | This study | STY34 | (*SEY6210*); *ist1Δ::TRP1; Mup1-pHluorin::NAT* |
| Strain, strain background (*S. cerevisiae, Mata*) | *vps60Δ* | This study | SBY465 | (*SEY6210.1*); *vps60Δ::hph; Mup1-pHluorin::Kan* |
| Strain, strain background (*S. cerevisiae, Mata*) | *vps20Δ* | PMID:24139821 | NBY42 | (*SEY6210.1*); *vps20::HIS3 MUP1-pHLUORIN::KANMX* |
| Strain, strain background (*S. cerevisiae, Mata*) | *vps24Δ vps2Δ* | This study | SBY05 | (*SEY6210.1*); *vps24Δ::HIS3; vps2Δ::Hph; MUP1-pHLUORIN::KANMX* |
| Strain, strain background (*S. cerevisiae, Mata*) | *vps60Δ* | PMID:18032584 | SMY24 | (*SEY6210.1*); *vps60Δ::TRP1* |

*Continued on next page*

*Appendix 1—key resources table continued*

| Reagent type (species) or resource | Designation | Source or reference | Identifiers | Additional information |
|---|---|---|---|---|
| Strain, strain background (*S. cerevisiae, Mata*) | *vps20Δ* | PMID:12194857 | MBY25 | (*SEY6210.1*); *vps20Δ::HIS3* |
| Strain, strain background (*S. cerevisiae, Mata*) | *snf7Δ* | PMID:12194857 | MBY24 | (*SEY6210.1*); *snf7Δ::HIS3* |
| Strain, strain background (*S. cerevisiae, Mata*) | *vps2Δ* | PMID:12194857 | MBY29 | (*SEY6210.1*); *vps2Δ::HIS3* |
| Strain, strain background (*S. cerevisiae, Mata*) | *did2Δ* | PMID:18032584 | SMY61 | (*SEY6210.1*); *did2Δ::TRP1* |
| Strain, strain background (*S. cerevisiae, Matα*) | *vps2Δ* | PMID:12194857 | MBY28 | (*SEY6210*); *vps2Δ::HIS3* |
| Strain, strain background (*S. cerevisiae, Matα*) | *did2Δ* | PMID:18032584 | SMY60 | (*SEY6210*); *did2Δ::TRP1* |
| Recombinant DNA reagent | Vector | PMID:2659436 | pRS416 | |
| Recombinant DNA reagent | Vector | PMID:2659436 | pRS414 | |
| Recombinant DNA reagent | Vector | PMID:2659436 | pRS415 | |
| Recombinant DNA reagent | pCM189 | PMID:9234672 | pCM189 | CMV promoter controlling Tet-off operator |
| Recombinant DNA reagent | pCM189 Vps24 (OE) | PMID:31246173 | pCM189 Vps24 | Progenitor: pRS 414 Vps24 (PMID:23063125), Vector: pCM189 (PMID:9234672), *S. cerevisiae* sequence |
| Recombinant DNA reagent | pCM189 Vps2 (OE) | PMID:31246173 | pCM189 Vps2 | Progenitor: pRS 415 Vps2 (PMID:23063125), Vector: pCM189 (PMID:9234672), *S. cerevisiae* sequence |
| Recombinant DNA reagent | Vps24 | PMID:31246173 | pRS 416 Vps24 | Progenitor: pRS 414 Vps24 (PMID:23063125), Vector: pRS416 (PMID:2659436), *S. cerevisiae* sequence |
| Recombinant DNA reagent | Vps2 | PMID:31246173 | pRS 416 Vps2 | Progenitor: pRS 415 Vps2 (PMID:23063125), Vector: pRS416 (PMID:2659436), *S. cerevisiae* sequence |
| Recombinant DNA reagent | Snf7 R52E | PMID:31246173 | pET28aH6SUMOSnf7R52E | Progenitor: pRS416 Snf7R52E (PMID:2659436), Vector: pET28aH6SUMO (PMID:26670543), *S. cerevisiae* sequence |

*Continued on next page*

*Appendix 1—key resources table continued*

| Reagent type (species) or resource | Designation | Source or reference | Identifiers | Additional information |
|---|---|---|---|---|
| Recombinant DNA reagent | Vps24 | PMID:31246173 | pET28aH6SUMOVps24 | Progenitor: pOPTVps24 (PMID:18786397), Vector: pET28aH6SUMO (PMID:26670543), *S. cerevisiae* protein sequence, DNA seq optimized for *Escherichia coli* |
| Recombinant DNA reagent | Vps2 | PMID:31246173 | pET28aH6SUMOVps2 | Progenitor: pOPTVps2 (PMID:18786397), Vector: pET28aH6SUMO (PMID:26670543) *S. cerevisiae* protein sequence, DNA seq optimized for *E. coli* |
| Recombinant DNA reagent | Vps60 | This study | pET23d + Vps60 | His6 tagged, *S. cerevisiae* sequence |
| Recombinant DNA reagent | Did2 | This study | pET23d + Did2 | His6 tagged, *S. cerevisiae* sequence |
| Recombinant DNA reagent | Vps2$^{RM}$ | This study | pRS 416 Vps2$^{RM}$ | Progenitor: obtained from random mutagenesis of pRS416 Vps2, transformed into NBY47 strain and selected for canvavanine resistance |
| Recombinant DNA reagent | Vps2* | This study | pRS 416 Vps2* | Vps2 N21K T28A E31K, Progenitor: pRS416 Vps2$^{RM}$ (promoter is the same as in the pRS416 Vps2$^{RM}$ plasmid) |
| Recombinant DNA reagent | pCM189 Vps20 | This study | pCM189 Vps20 | CMV promoter controlling Tet-off operator |
| Recombinant DNA reagent | pCM189 Vps60 | This study | pCM189 Vps60 | CMV promoter controlling Tet-off operator |
| Recombinant DNA reagent | pCM189 Ist1 | This study | pCM189 Ist1 | CMV promoter controlling Tet-off operator |
| Recombinant DNA reagent | pCM189 Did2 | This study | pCM189 Did2 | CMV promoter controlling Tet-off operator |
| Recombinant DNA reagent | Vps2 N21K | This study | pRS 416 Vps2 N21K | WT promoter |
| Recombinant DNA reagent | Vps2 E31K | This study | pRS 416 Vps2 E31K | WT promoter |
| Recombinant DNA reagent | Vps2 T28A | This study | pRS 416 Vps2 T28A | WT promoter |
| Recombinant DNA reagent | Vps2 N21K E31K | This study | pRS 416 Vps2 N21K E31K | WT promoter |
| Recombinant DNA reagent | Vps2 N21K T28A E31K | This study | pRS 416 Vps2 N21K T28A E31K | WT promoter |
| Recombinant DNA reagent | Vps24WT | PMID:23063125 | pRS 414 Vps24 | *S. cerevisiae* sequence |
| Recombinant DNA reagent | Vps24$^{E114K}$ | This study | pCM189 Vps24 E114K | *Figure 3* |
| Recombinant DNA reagent | Vps24$^{E114K}$ - Vps2 | This study | pCM189 Vps24 E114K - Vps2 | *Figure 3* |

*Continued on next page*

*Appendix 1—key resources table continued*

| Reagent type (species) or resource | Designation | Source or reference | Identifiers | Additional information |
|---|---|---|---|---|
| Recombinant DNA reagent | Vps24-(Vps2 α5/MIM) | This study | pCM189 Vps24 -(Vps2 α5/MIM) | *Figure 3* |
| Recombinant DNA reagent | Vps24$^{E114K}$-(Vps2 α5/MIM) | This study | pCM189 Vps24-E114K-(Vps2 α5/MIM) | *Figure 3* |
| Recombinant DNA reagent | Vps24$^{E114K}$-(Vps2 MIM) | This study | pCM189 Vps24-E114K-(Vps2 MIM) | *Figure 3* |
| Recombinant DNA reagent | vps24(1-56)-vps2 | This study | pRS414 vps24(1-56)-vps2 | *Figure 3—figure supplement 1* |
| Recombinant DNA reagent | vps24(1-106)-vps2 | This study | pRS414 vps24(1-106)-vps2 | *Figure 3—figure supplement 1* |
| Recombinant DNA reagent | vps24(1-119)-vps2 | This study | pRS414 vps24(1-119)-vps2 | *Figure 3—figure supplement 1* |
| Recombinant DNA reagent | vps24(1-152)-vps2 | This study | pRS414 vps24(1-152)-vps2 | *Figure 3—figure supplement 1* |
| Recombinant DNA reagent | vps24-(vps2 α5/MIM) | This study | pRS414 vps24-(vps2 α5/MIM) | *Figure 3—figure supplement 1* |
| Recombinant DNA reagent | vps24$^{E114K}$-(vps2 α5/MIM) | This study | pRS414 vps24$^{E114K}$-(vps2 α5/MIM) | *Figure 3—figure supplement 1* |
| Recombinant DNA reagent | Vps2(1-56)-Vps24 | This study | 416-Vps2(1-56)-Vps24(57-224) | *Figure 3—figure supplement 3* |
| Recombinant DNA reagent | Vps2(1-99)-Vps24 | This study | 416-Vps2(1-99)-Vps24(101-224) | *Figure 3—figure supplement 3* |
| Recombinant DNA reagent | Vps2(1-119)-Vps24 | This study | 416-Vps2(1-119)-Vps24(120-224) | *Figure 3—figure supplement 3* |
| Recombinant DNA reagent | Vps2(1-164)-Vps24 | This study | 416-Vps2(1-164)-Vps24(164-224) | *Figure 3—figure supplement 3* |
| Recombinant DNA reagent | Vps2(1-217)-Vps24 | This study | 416-Vps2(1-217)-Vps24(213-224) | *Figure 3—figure supplement 3* |
| Recombinant DNA reagent | Vps2(1-56)-Vps24 | This study | pCM189 Vps2(1-56)-Vps24(57–224) | *Figure 3—figure supplement 3* |
| Recombinant DNA reagent | Vps2(1-99)-Vps24 | This study | pCM189 Vps2(1-99)-Vps24(101-224) | *Figure 3—figure supplement 3* |
| Recombinant DNA reagent | Vps2(1-119)-Vps24 | This study | pCM189 Vps2(1-119)-Vps24(120-224) | *Figure 3—figure supplement 3* |
| Recombinant DNA reagent | Vps2(1-164)-Vps24 | This study | pCM189 Vps2(1-164)-Vps24(164-224) | *Figure 3—figure supplement 3* |
| Recombinant DNA reagent | Vps2(1-217)-Vps24 | This study | pCM189 Vps2(1-217)-Vps24(213-224) | *Figure 3—figure supplement 3* |
| Recombinant DNA reagent | Vps24 NG (V2 H5 MIM) | This study | pCM189 Vps24 N99G N103G (Vps2 H5 MIM) | *Figure 4* |
| Recombinant DNA reagent | Vps24 NA (V2 H5 MIM) | This study | pCM189 Vps24 N99A N103A (Vps2 H5 MIM) | *Figure 4* |
| Recombinant DNA reagent | Vps2 (α2/α3–24) | This study | pCM189 Vps2 (α2/α3–24) | *Figure 4* |
| Recombinant DNA reagent | *Vps60-GFP* | PMID:18032584 | pSM11; pRS415-Vps60-GFP | |
| Recombinant DNA reagent | *VPS60* | This study | pRS416-Vps60 | *Figure 6* |
| Recombinant DNA reagent | *snf7(1-59)* | This study | pRS416-snf7(1-59)-Vps60 | *Figure 6* |

*Continued on next page*

*Appendix 1—key resources table continued*

| Reagent type (species) or resource | Designation | Source or reference | Identifiers | Additional information |
|---|---|---|---|---|
| Recombinant DNA reagent | *snf7(1-121)* | This study | pRS416-snf7(1-121)-Vps60 | *Figure 6* |
| Recombinant DNA reagent | *snf7(1-155)* | This study | pRS416-snf7(1-155)-Vps60 | *Figure 6* |
| Recombinant DNA reagent | Snf7 R52E | PMID:31246173 | pET28aH6SUMOSnf7R52E | Progenitor: pRS416 Snf7R52E (PMID:2659436), Vector: pET28aH6SUMO (PMID:26670543), *S. cerevisiae* sequence |
| Recombinant DNA reagent | Snf7 (vps2α5MIM) | This study | 416-Snf7(vps2α5MIM) | *Figure 3—figure supplement 2* |
| Recombinant DNA reagent | Snf7*** (vps2α5MIM) | This study | 416-Snf7***(vps2α5MIM) | *Figure 3—figure supplement 2* |
| Recombinant DNA reagent | Snf7(vps2MIM) | This study | 416-Snf7(vps2MIM) | *Figure 3—figure supplement 2* |
| Recombinant DNA reagent | Snf7*** (vps2MIM) | This study | 416-Snf7***(vps2MIM) | *Figure 3—figure supplement 2* |
| Recombinant DNA reagent | Snf7(Vps2α5) | This study | 416-Snf7(Vps2α5) | *Figure 3—figure supplement 2* |
| Recombinant DNA reagent | Snf7*** (Vps2α5) | This study | 416-Snf7***(Vps2α5) | *Figure 3—figure supplement 2* |
| Antibody | anti-Vps2 (Rabbit polyclonal) | PMID:24711499 | anti-Vps2 Ab, *S. cerevisiae* Vps2 | (1:500) |
| Antibody | anti-GFP (Rabbit polyclonal) | Torrey Pines Biolabs | (Torrey Pines Biolabs Cat# TP401, RRID:AB_2313770) | (1:2500) |
| Antibody | anti-G6PDH (Rabbit polyclonal) | Sigma-Aldrich | (Sigma-Aldrich Cat# A9521, RRID:AB_258454) | (1: 10000) |
| Antibody | anti-PGK (Mouse monoclonal) | Thermo Fisher | (Thermo Fisher Scientific Cat# 459250, RRID:AB_2532235) | (1:4000) |
| Antibody | anti-Mouse IRdye 800/680 (Goat polyclonal) | LI-COR Biosciences | LI-COR Biosciences Cat# 926–32210, RRID:AB_621842 | (1:10,000) |
| Antibody | anti-Rabbit IRdye 800/680 (Rabbit polyclonal) | LI-COR Biosciences | LI-COR Biosciences Cat# 926–32211,RRID:AB_621843 | (1:10,000) |
| Antibody | anti-Snf7 (Rabbit polyclonal) | PMID:9606181 | anti-Snf7 Ab, *S. cerevisiae* Snf7 | (1:10000) |
| Antibody | anti-Vps24 (Rabbit polyclonal) | PMID:9606181 | anti-Vps24 Ab, *S. cerevisiae* Vps24 | (1:1000) |
| Software | Odyssey, Image Studio Lite | LI-COR Biosciences | (Image Studio Lite, RRID:SCR_013715) | |
| Software | Mafft | http://mafft.cbrc.jp/alignment/server/ | (MAFFT, RRID:SCR_011811) | |

*Continued on next page*

*Appendix 1—key resources table continued*

| Reagent type (species) or resource | Designation | Source or reference | Identifiers | Additional information |
|---|---|---|---|---|
| Software | Jalview | PMID:9151095 | (Jalview, RRID:SCR_006459) | |
| Software | Modeller | https://salilab.org/modeller/download_installation.html | (MODELLER, RRID:SCR_008395) | |
| Software | UCSF Chimera | PMID:15264254 | (UCSF Chimera, RRID:SCR_004097) | |
| Software | Fiji (ImageJ) | PMID:22743772 | (Fiji, RRID:SCR_002285) | |
| Other | Canavanine | Sigma-Aldrich | L-Canavanine sulfate, Cat.# G8772 | Stock solution 5–10 mg/mL, made in water |
| Other | Cobalt resin | Clontech | TALON Metal Affinity Resin, Cat.# 635502 | |
| Other | HiTrap Q column | GE Healthcare | HiTrap Q Sepharose FF, Cat.# 17-5053-01 | |
| Other | SD200increase | GE Healthcare | Superdex 200 Increase 10/300 GL, Cat.# 28990944 | |
| Other | EM grids | Electron Microscopy Sciences | Formvar Carbon Coated 200 Mesh Copper Grids, Cat.# FCF200-Cu | |
| Other | Ammonium molybdate | Sigma-Aldrich | Ammonium molybdate 99.98%, Cat.# 277908 | Made 2% (w/v) in water |
| Other | POPC | Avanti Polar Lipids | 16:0-18:1 PC (POPC), Cat.# 850457C | |
| Other | POPS | Avanti Polar Lipids | 16:0-18:1 PC (POPC), Cat.# 840034 | |
| Other | PI3P | Avanti Polar Lipids | 18:1 PI(3)P, Cat.# 850150 | Dissolved in 10% methanol, 90% chloroform, to make 0.25–1 mg/mL of solution |
| Other | Doxycycline | Sigma-Aldrich | Doxycycline hyclate, Cat.# D9891 | Stock solution 5 mg/mL, made in ethanol |

