## [Decision Letter]

**Acceptance summary:**

Different ESCRT-III subunits share sequence similarity but have been characterized in distinct conformations and perform distinct roles in the polymerization process that is central to ESCRT membrane fission pathways. Here it is shown that mutations in one ESCRT-III subunit can compensate for loss of a different subunit by encoding the specialized roles of both subunits within one polypeptide. Specifically, the work provides new insight into the interaction of key ESCRT-III members Vps2 and Vps24. They also test functional similarities within sequence-defined subgroups of ESCRT-III proteins. Combined with previous studies, the authors move toward defining minimal requirements for a functional ESCRT-III complex. This provides insights into why certain systems function with only a subset of ESCRT-III proteins, and forms a basis for understanding how the system has been elaborated for specific organelle biogenesis.

**Decision letter after peer review:**

Thank you for submitting your article "Design Principles of the ESCRT-III Vps24-Vps2 Module" for consideration by *eLife*. Your article has been reviewed by 3 peer reviewers, one of whom is a member of our Board of Reviewing Editors, and the evaluation has been overseen by Suzanne Pfeffer as the Senior Editor. The following individual involved in review of your submission has agreed to reveal their identity: Christopher P Hill (Reviewer #3).

Essential revisions:

The authors should soften some of their claims and amplify certain discussion points, as follows:

1. The authors show that 8-fold over expression is necessary to rescue Mup1 sorting to an extent of 40%. The authors hypothesize that over expression of Vps2 can rescue Vps24 deletion because Vps2 may have a lower affinity for Snf7 than Vps24. This is in agreement with data on mammalian homologues which showed that indeed CHMP3 binds with 10x higher affinity to CHMP4B than CHMP2A (Effantin et al., 2012). This should be included in the discussion, since the function of yeast and mammalian core ESCRT-III proteins is most likely not different.

2. The authors designed several chimeric Vps24/Vps2 constructs and show that some of the Vps24 chimera including Vps2 helix 5 and the MIM are fully functional in Mup1 sorting in δ Vps24 cells, but lack the ability to functionally replace Vps2 in Vps2 δ cells. Are the chimeras in the closed conformation in the cytosol? They are probably activated more easily and possibly prematurely?

3. The authors show that Vps24 E114K can form some kind of polymers in the presence of Vps2 in vitro while no polymerization is observed for wt Vps24 at 1 µM. Does wt Vps24 polymerize at higher concentrations in this assay?

4. While the conclusion that E114K shifts the equilibrium to the open state is plausible, there is no evidence provided that this mimics Vps2 as stated. If so, Vps24 E114K should form the same polymers as shown in figure 4 supp 1 in the absence of Vps2 and spiral formation with snf7 should not require Vps2.

5. The speculation in the Results section that Vps24 may not extend its helices 2 and 3 in an activated form due to potential helix breaking Asn residues in the linker region is not backed up by any data and should be moved to the discussion.

6. The proposal that Vps2-Vps24 heteropolymers are formed by interactions along helices 2 and 3 is not supported by data presented in the manuscript. The authors should use recombinant proteins to test their mutants in biophysical interaction studies or soften their statement.

*Reviewer #1 (Recommendations for the authors):*

The model shown in Figure 7 needs to more clearly label the proposed, distinct roles for the Snf7 and Vps24 subgroups.

---

## [Author Response]

Essential revisions:The authors should soften some of their claims and amplify certain discussion points, as follows:

We have made several changes to the manuscript – amplified some concepts, softened some interpretations and added new data. In particular we have softened our interpretations of the predicted conformational changes induced by mutations in Vps24-Vps2. A point-by-point response to the reviewers’ comments follows.

1. The authors show that 8-fold over expression is necessary to rescue Mup1 sorting to an extent of 40%. The authors hypothesize that over expression of Vps2 can rescue Vps24 deletion because Vps2 may have a lower affinity for Snf7 than Vps24. This is in agreement with data on mammalian homologues which showed that indeed CHMP3 binds with 10x higher affinity to CHMP4B than CHMP2A (Effantin et al., 2012). This should be included in the discussion, since the function of yeast and mammalian core ESCRT-III proteins is most likely not different.

We have now cited and discussed the Effantin et al., 2013 paper, as suggested:

“These ideas are also consistent with previous binding-constant measurements with mammalian Vps2 (CHMP2A) and Vps24 (CHMP3), which showed that CHMP3 possesses a 16-fold tighter affinity than CHMP2A to mammalian Snf7 (CHMP4A) (Effantin et al., 2013).”

2. The authors designed several chimeric Vps24/Vps2 constructs and show that some of the Vps24 chimera including Vps2 helix 5 and the MIM are fully functional in Mup1 sorting in δ Vps24 cells, but lack the ability to functionally replace Vps2 in Vps2 δ cells. Are the chimeras in the closed conformation in the cytosol? They are probably activated more easily and possibly prematurely?

We attempted, but currently don’t have a clear assay to quantitatively distinguish open and closed conformations of Vps24. Therefore, we cannot definitively say that the chimeras of Vps24-Vps2 (Vps2 helix5 and the MIM) are either in closed or open conformations.

However, we believe that these chimeras may not be activated easily and prematurely, since this would probably cause a defect in function. The chimeras are fully functional in vps24∆, as the reviewer noted.

In a new set of data, we also present evidence that when incorporated into Snf7, the helices 5 and MIM motifs of Vps2 make this chimeric Snf7 dysfunctional (lines 185-199, Figure 3 – Supp. 3). These data are consistent with the reviewers’ interpretation that premature recruitment of Vps4 to ESCRT-III filaments is presumably dysfunctional. However, inclusion of these motifs to Vps24 most likely does not prematurely disassemble ESCRT-III filaments, hence they remain functional. Additionally, these Snf7-Vps2 chimeras do not rescue the absence of Vps24 or Vps2, suggesting that merely including the Vps4 binding site on Snf7 is not enough for ESCRT-III function.

The larger point behind this set of analyses is that there are additional functions of Vps24-Vps2 beyond just recruitment of the AAA+ ATPase Vps4. Since we extensively analyzed the lateral association of Vps24-Vps2 to Snf7 in our previous manuscript (Banjade et al., *eLife* 2019), we ascribe these additional functions to lateral polymerization of Vps24-Vps2 on the Snf7 filament.

3. The authors show that Vps24 E114K can form some kind of polymers in the presence of Vps2 in vitro while no polymerization is observed for wt Vps24 at 1 µM. Does wt Vps24 polymerize at higher concentrations in this assay?

In our assay, we don’t observe polymers with higher concentrations (15 µM of Vps24 and 15 µM of Vps2), as we start observing amorphous aggregates on the grids (Figure 4 Supp 1 C). We do refer to other manuscripts in the past (Ghazi-Tabatabai et al., 2008), who have observed similar linear polymers of Vps24 at higher concentrations (>300 µM) and longer incubation times (overnight, as opposed to 10 minutes on grids in our case). So we believe that the ESCRT-III proteins Vps24 and Vps2 are able to form copolymers with a similar structure that is enhanced when these “activating” mutations are included.

4. While the conclusion that E114K shifts the equilibrium to the open state is plausible, there is no evidence provided that this mimics Vps2 as stated. If so, Vps24 E114K should form the same polymers as shown in figure 4 supp 1 in the absence of Vps2 and spiral formation with snf7 should not require Vps2.

We agree with this interpretation from the in vitro assays, and have appropriately changed the language in the manuscript. We now describe the effect of the E114K protein to “enhance” association with existing Vps2. We hypothesize that this enhanced association to Vps2 occurs due to an “activation” process whereby Vps24 adopts a higher population of an open (or a semi-open) conformation, and have changed the language to reflect this interpretation. As an aside, we do note that Snf7 and Vps24 do form helices at higher concentrations without Vps2, as we showed in Banjade et al., *eLife* 2019.

5. The speculation in the Results section that Vps24 may not extend its helices 2 and 3 in an activated form due to potential helix breaking Asn residues in the linker region is not backed up by any data and should be moved to the discussion.

We have now moved this speculation to the Discussion section, and also added the following sentence when describing the data regarding the mutations in the potential helix-breaking Asn residues:

“We note that these data are indicative of mutations that control the conformations of the proteins. However, further biophysical analyses will be required for definitive evidence of this conformational flexibility.”

6. The proposal that Vps2-Vps24 heteropolymers are formed by interactions along helices 2 and 3 is not supported by data presented in the manuscript. The authors should use recombinant proteins to test their mutants in biophysical interaction studies or soften their statement.

We have now moved this hypothesis to the Discussion section and softened the statement. We note that this hypothesis is also based on recent data from the Teis lab (Sprenger et al., Biorxiv), which are consistent with the idea that the helices 2 and 3 are involved in longitudinal interactions of Vps24-Vps2. We believe that further biophysical analyses would be appropriate in a future manuscript.

Reviewer #1 (Recommendations for the authors):The model shown in Figure 7 needs to more clearly label the proposed, distinct roles for the Snf7 and Vps24 subgroups.

We have modified the figure to specify the roles of Snf7 and Vps24/Vps2.